# Low Expression of MATR3 Is Associated with Poor Survival in Clear Cell Renal Cell Carcinoma

**DOI:** 10.3390/biomedicines11020326

**Published:** 2023-01-24

**Authors:** Justyna Durślewicz, Anna Klimaszewska-Wiśniewska, Paulina Antosik, Dariusz Grzanka

**Affiliations:** Department of Clinical Pathomorphology, Faculty of Medicine, Collegium Medicum in Bydgoszcz, Nicolaus Copernicus University in Torun, 85-094 Bydgoszcz, Poland

**Keywords:** clear cell renal cell carcinoma, ccRCC, MATR3, prognostic biomarkers, nuclear matrix protein

## Abstract

Matrin 3 (MATR3) is one of the most abundant inner nuclear matrix proteins involved in multiple nuclear processes. However, to date, the biological role and prognostic relevance of MATR3 in human cancers still need to be explored. Therefore, the present study aimed to examine the expression levels and prognostic significance of MATR3 in clear cell renal cell carcinoma (ccRCC) patients. We assessed MATR3 immunohistochemical staining and RNA-seq data from publicly available data sets, and the results were analyzed with reference to clinicopathological characteristics and the overall survival of patients. Furthermore, the protein–protein interaction (PPI) network for *MATR3* and its neighbors was constructed, functionally annotated, and screened for survival-related genes. MATR3 protein and mRNA levels were lower in tumor tissues compared to control tissues. Lower MATR3 protein (HR 2.36, 95%CI 1.41–3.97; *p* = 0.001) and mRNA (HR 2.01, 95%CI 1.46–2.75; *p* < 0.0001) expression levels were found to be a significant independent adverse prognostic factor for the patient’s overall survival (OS). Moreover, of the candidate genes, the *MRPL23* gene was identified as being the most predictive of OS, and combined *MRPL23/MATR3* expression status predicted patient survival better than looking at each marker individually (HR 3.15, 95%CI 2.05–4.83; *p* < 0.0001). In conclusion, the results from the present investigation warrant further research into the biological and prognostic value of MATR3 and MRPL23 in ccRCC patients.

## 1. Introduction

Renal cell carcinoma (RCC) is one of the most malignant urinary tumors with respect to mortality, and its morbidity has gradually increased in recent years. Clear cell renal cell carcinoma (ccRCC) is the most common histological subtype of RCC, accounting for > 80% of all cases [1,2]. Considering the lack of characteristic early clinical manifestations and reliable clinical diagnostic biomarkers, approximately 30% of patients with ccRCC already have metastases at diagnosis [3]. Metastatic ccRCC has a poor prognosis, with a 5-year overall survival (OS) rate of approximately 10% and a median survival of about 13 months [4]. Therefore, identifying new diagnostic markers, prognostic factors, and therapeutic targets is essential to enhance the survival of patients with ccRCC.

Matrin3 (MATR3), one of the most abundant innernuclear matrix proteins, is involved in many processes, including mRNA assembly/stabilization, nuclear retention of hyper-edited RNAs, RNA splicing, and viral RNA regulation and the DNA damage response [5,6]. MATR3 is a DNA- and RNA-binding protein, and genetic changes in the *MATR3* gene play a role in the biology of neurodegenerative disorders [7,8]. In cancer, *MATR3* was recognized as one of the five genes deleted from chromosome 5 of primary, metastasized, and xenografted human basal-like breast cancer (BLBC), which allowed for a proposal of a tumor suppressive function of this gene [9]. However, the biological role and prognostic relevance of MATR3 in human cancers remain largely under explored.

This paper hypothesizes that MATR3 expression has prognostic implications in ccRCC. To test this thesis, we analyzed MATR3 protein and mRNA expression in ccRCC using our own and public datasets, respectively, focusing on its effect on clinicopathological features and overall survival (OS) status in ccRCC. Furthermore, the protein–protein interaction (PPI) network for *MATR3* and its neighbors was constructed and functionally annotated, allowing insight into the roles they could potentially play in ccRCC. Lastly, survival-related genes were identified and then verified for their combined prognostic value with *MATR3*. 

## 2. Materials and Methods

### 2.1. Patients and Tissue Material

A total of 132 patients operated on at the Department of Urology and Andrology, Antoni Jurasz University Hospital No. 1 in Bydgoszcz, Poland, were screened for inclusion in the study in the first round. To avoid excessive investigation complexity, the cohort included ccRCC, while all other histological types were eliminated from the research series. Two independent pathologists conducted histopathological examinations at the Department of Clinical Pathomorphology, Collegium Medicum in Bydgoszcz of the Nicolaus Copernicus University in Torun. The samples were excluded from subsequent analysis if the quality of the material collected was unacceptable. Finally, formalin-fixed, paraffin-embedded (FFPE) tissue specimens were obtained from 107 patients with ccRCC (shown in Appendix A). The study group comprised 75 males and 32 females with a median age of 64 years (range: 42–83). Twenty-six patients had well-differentiated, 68 patients moderately differentiated, 12 patients poorly differentiated, and one patient had undifferentiated ccRCC. Postsurgical survival data were available for all patients. The median follow-up time was 1282 days. The detailed characterization of the study group is shown in Appendix A.

The study was approved by the Ethics Committee of Collegium Medicum in Bydgoszcz of the Nicolaus Copernicus University in Torun (approval number KB 253/2018).

### 2.2. Immunohistochemistry on Tissue Macroarrays

Immunohistochemical (IHC) staining was carried out on tissue macromatricescomposed of representative tumor areas. Five different large tissue fragments from donor paraffin blocks were present in one recipient block. The sections (4 μm thick) were cut from each tissue macroarray block and placed on high-adhesive glass slides (SuperFrost Plus; Menzel-Glaser, Braunschweig, Germany). The sections were subjected to IHC staining according to the previously described protocol [10,11]. Immunohistochemistry was carried out by BenchMark® Ultra automatic staining machine (Roche Diagnostics/Ventana Medical Systems, Tucson, AZ, USA) with Ventana UltraView DAB Detection Kit (Ventana Medical Systems). MATR3 was detected by using anti-MATR3 rabbit polyclonal antibody (cat. no: HPA036565, Sigma-Aldrich, St. Louis, MO, USA) in a 1:300 dilution for 32 min.

### 2.3. Evaluation of Immunohistochemistry Staining

The pathologist evaluated slides in a blinded fashion, using an Olympus BX53 (Olympus, Tokyo, Japan) light microscope at 20× original objective magnification. The immunoexpression was analyzed based on the H-score system. The scoring system for analyzed proteins was determined by multiplying staining intensity (0–3) and the percentage of positively stained cells. The final score, ranging from 0 to 300, was dichotomized into negative (low) and positive (high) expression based on a defined discriminatory cutoff established by the Evaluate Cutpoints software [12]. The cutoff values for low and high MATR3 were <15 and ≥15, respectively.

### 2.4. Analysis of the Database

Apart from the protein level, the mRNA expression levels of the targeted genes were compared using public data. *MATR3* mRNA expression between ccRCC and “match TCGA normal and GTEx data” was compared using the public web server GEPIA2 (Gene Expression Profiling Interactive Analysis; http://gepia.cancer-pku.cn/, accessed on 11 April 2022). Furthermore, gene expression, clinicopathological, and survival data for a cohort of 475 ccRCC patients were obtained from the UCSC Xena Browser (http://xena.ucsc.edu/, accessed on 8 April 2022). RNA-sequencing data for *MATR3* were normalized via DESeq2 normalization. The detailed characterization of the TCGA cohort is shown in Appendix A. The data were categorized into low-level (<10.77) and high-level (≥10.77) MATR3 expression groups according to the cutoff point established using the Evaluate Cutpoints software [12].

### 2.5. PPI Network Construction and Analysis

The list of genes positively and negatively correlated with *MATR3* in ccRCC was downloaded from the UALCAN database (http://ualcan.path.uab.edu/, accessed on 28 April 2022) [13]. The top 50 genes from each group were selected for further analyses based on their respective Pearson CC values. The Search Tool for the Retrieval of Interacting Genes Database (STRING) and Cytoscape software version 3.8.2 was then used to construct and visualize the zero-order (seed proteins only) protein–protein interaction network (PPI). Topological parameters of the PPI network, including, e.g., the number of nodes and edges, average number of neighbors, characteristic path length, clustering coefficient, network degree, betweenness centrality, and closeness centrality were obtained from the Network analyzer plugin (version 4.4.8) in the Cytoscape software. Furthermore, using the ClueGO (version 2.5.8) plugin, GO terms, Reactome Pathways, and KEGG enrichment analyses were performed, with a threshold of *p* ≤ 0.05 based on a two-sided hypergeometric test (kappa score 0.4) and the Benjamini–Hochberg correction. The hub genes were determined from the PPI network using five topological analysis methods of the CytoHubba plugin (version 0.1). The Molecular Complex Detection (MCODE) clustering algorithm (version 2.0.0) was utilized to select the core modules in the PPI network according to the clustering score using the following criteria: degree cutoff = 2, max. depth = 100, k-core = 2, haircut = yes, and node score cutoff = 0.2. The most significant subnet works were functionally annotated using the *DAVID Functional Annotation Clustering* Tool (The Database for Annotation, Visualization and Integrated Discovery version 6.8, DAVID; https://david.ncifcrf.gov, accessed on 28 April 2022). A combined view was obtained from the DAVID-defined defaults and Reactome Pathways. 

### 2.6. Survival Analysis of Candidate Genes

Overall survival curves for genes positively and negatively correlated with *MATR3* in ccRCC were obtained from the UALCAN database. To identify the essential\ genes related to OS of ccRCC patients, we focused on the top module genes (clusters 1 and 2) along with those with the highest degree and betweenness centralities. We downloaded the TCGA gene expression data via the UCSC Xena database (except for *TCEB2*, for which the expression data was unavailable). We performed a Kaplan–Meier and univariate Cox analysis with a cutoff criterion of *p* < 0.05. After that, variables significant in these analyses were introduced into the multivariate Cox model in a backward stepwise fashion to shrink the OS gene range. Two genes remained significant in the model, and the combined prognostic value of these genes was evaluated by the Kaplan–Meier analysis is as well as by univariate and multivariate Cox models. Cases with *MATR3*-high and *MRPL23*-low coexpression were analyzed against those with the opposite expression pattern (*MATR3*-low/*MRPL23*-high), whereas ‘others’ defined cases expressing either both genes at high levels, or both at low levels (i.e., *MATR3*-high/*MRPL23*-high or *MATR3*-low/*MRPL23*-low). The UALCAN web portal and TNMplot.com (https://www.tnmplot.com, accessed on 4 May 2022) were utilized to evaluate the expression profile of the *MRPL23* gene in ccRCC and adjacent normal tissue based on the TCGA RNA-seq data.

### 2.7. Statistical Analysis

Statistical analyses were conducted using SPSS software packages version 26.0 (IBM Corporation, Armonk, NY, USA) and GraphPad Prism version 8.0 (GraphPad Software, San Diego, CA, USA) utilizing the Shapiro–Wilk test, Mann–Whitney test, Fisher test, and Chi-squared test. Overall survival was plotted using Kaplan–Meier plots. Univariate and multivariate survival analyses were performed using Cox proportional hazards regression. The hazard ratios (HRs) and 95% confidence intervals (95% CIs) were also calculated. Multivariate Cox proportional hazard analyses of our own cohort and the TCGA cohort were adjusted for all univariate (*p* ≤ 0.06) predictors of survival, including sex (male vs. female), age (<65 years vs.≥65 years for our cohort; ≤60 years vs.>60 years for the TCGA cohort), grade (G1,G2 vs. G3,G4), pN and cN status (N0 vs. N1), and AJCC pathological stage (stage I, II vs. stage III, IV for TCGA cohort). *p*-values <0.05 were considered statistically significant.

## 3. Results

### 3.1. Immunoexpression of MATR3 in ccRCC and Normal Adjacent Tissue—Clinicopathological Associations

Immunohistochemical staining of MATR3 was detected in the nuclear compartments of ccRCC and normal renal parenchyma tissues. Representative images are presented in Figure 1. Twenty samples (18.69%) of tumor tissue were characterized by low MATR3 expression and 87 (81.31%) by high. MATR3 expression was decreased in ccRCC tissues compared with healthy margin tissue (*p* < 0.0001; Figure 2A). The expression status of MATR3 was not associated with clinicopathological features in our cohort (Table 1).

### 3.2. Prognostic Value of MATR3 Immunoexpression in Predicting the Overall Survival of ccRCC Patients

In our study, patients with low MATR3 protein levels had a worse OS than patients with high MATR3 protein levels (median OS: 325 days vs 1606 days; *p* < 0.0001; Figure 2B). The univariate Cox analysis revealed that low MATR3 protein expression predicted an unfavorable OS (HR 2.59, 95%CI 1.57–4.27; *p* < 0.0001; Table 2). In the multivariate Cox proportional hazards model, protein expression of MATR3 was an independent prognostic factor for OS, after adjusting for sex, age, grade, and cN status (HR 2.36,95%CI 1.41–3.97; *p* = 0.001; Table 2).

### 3.3. MATR3 mRNA Expression in Tumor and Normal Adjacent Tissue Derived from Public Datasets—Clinicopathological Associations

The expression of *MATR3* mRNA from the GEPIA2 (Figure 2C) and UALCAN (Appendix A) databases was lower in ccRCC tissues compared to normal renal tissues. Based on the established cutoff, low *MATR3* mRNA expression from the TCGA database was observed for 199 cases (41.89%). The expression status of *MATR3* in the TCGA cohort was associated with age (*p* = 0.03), pT status (*p* = 0.01), and pN status (*p* = 0.01). Low expression of *MATR3* mRNA was found more frequently in younger than in older people (*p* = 0.03); however, the association with age was confirmed neither in the continuous data analysis (Appendix A), nor in the additional analysis based on the UALCAN database (Appendix A). The ratio of *MATR3* overexpression was also significantly higher in patients with pT1 ccRCC than in those with pT2, pT3, and pT4 tumors (*p* = 0.01). Moreover, positive mRNA expression of *MATR3* was more common in patients without cancer cells in lymph nodes than in those with lymph node metastases (*p* = 0.01). The relationship between *MATR3* expression and ccRCC clinicopathological features for the TCGA cohort is summarized in Table 3.

### 3.4. Prognostic Value of MATR3 mRNA Expression in Predicting the Overall Survival of ccRCC Patients from Public Dataset

From the in silico analysis of the TCGA data, patients with low MATR3 mRNA levels had worse OS than patients with high MATR3 mRNA levels (median OS:1913 days vs. 3615 days; *p* < 0.0001; Figure 2D). The univariate Cox analysis revealed that low MATR3 predicted an unfavorable OS (HR 1.97, 95%CI 1.44–2.70; *p* < 0.0001; Table 4). When examined in the multivariate analysis, MATR3 remained the independent prognostic factor in terms of OS (HR 2.01,95%CI 1.46–2.75; *p* < 0.0001; Table 4).

### 3.5. PPI Network Construction and Analysis

The list of genes positively and negatively correlated with *MATR3* in ccRCC was downloaded from the UALCAN database. To assess the protein–protein connections among *MATR3* and coexpressed genes, we utilized the STRING online database to compute the protein interactions and plotted them using Cytoscape. The PPI network was initially constructed by importing the 101 genes (i.e., *MATR3* along with 50 top *MATR3*-positively correlated genes and 50 top *MATR3*-negatively correlated genes, Appendix A) into the STRING. Next, the PPI network, composed of 70 nodes and 85 edges, was depicted using Cytoscape, as shown in Figure 3A. Functional enrichment analysis was then applied to this set of proteins to provide insights into the role they potentially play in ccRCC. The enriched functional terms related to the queried genes mainly included the exonucleolytic nuclear-transcribed mRNA catabolic process involved in deadenylation-dependent decay, prenyltransferase activity, mitochondrial translation initiation, DNA damage bypass, RNA polymerase II pre-transcription events, and negative regulation of ubiquitination (Figure 4A,B).

Furthermore, the Network Analyzer and CytoHubba were used to score and rank the nodes by network features. The top 10 nodes with the highest degree, betweenness centrality, closeness centrality, bottleneck scores, and clustering coefficient are listed in Figure 3B, C, D, E, and F, respectively. In addition, by performing gene module analysis using the MCODE plugin in the Cytoscape software, six cluster subnetworks were identified from the PPI network (Figure 3G); those with a clustering score above three (the top two clusters; Figure 3H) were subjected to functional annotation using the *DAVID Functional Annotation Clustering* Tool. Cluster 1, consisting of 5 genes being negatively correlated with *MATR3* expression, was implicated in protein metabolism, including cytoplasmic translation and mitochondrial translation (Appendix A). Cluster 2 had enriched expression of genes positively correlated with *MATR3* and associated with intracellular protein transport (Appendix A).

### 3.6. Overall Survival-Related Gene Screening

According to the UALCAN database, the vast majority of *MATR3*-correlated genes were significantly associated with OS in ccRCC patients. We next focused on the top module genes (cluster 1 and 2) along with those with the highest degree (Figure 3B) and betweenness centralities (Figure 3C), i.e., the 2 measures widely used in network theory, as the fundamental parameters for evaluating the nodes in different PPIs associated with diseases [14]. Kaplan–Meier and univariate Cox analysis of ccRCC patients enrolled from the TCGA via the UCSC Xena database revealed that of the 19 genes, 15 genes significantly affected the OS (Appendix A). Among candidate genes positively correlated with *MATR3*, six genes were found to be significantly associated with better survival when overexpressed in ccRCC, including *TBC1D15*, *KHDRBS1*, *STAG2*, *G3BP1*, *CDC73*, and *CHM* (*p* < 0.05; Appendix A)*,* while *VPS26A*, *RAB6A*, *STX12*, and *PUM2* tended to be related to better OS (*p* < 0.2; Appendix A). Genes negatively correlated with *MATR3*, including *GADD45GIP1*, *MRPL23 (RPL23L)*, *MRPL41*, *MRPS15*, *RPS15*, *NDUFS8*, *C19orf53*, *BLOC1S1*, and *SERF2* were all significantly associated with poor OS when overexpressed in ccRCC (Appendix A). Compared to other candidate genes, Mitochondrial Ribosomal Protein L23 (*MRPL23*) had the highest prognostic hazard ratio value and the smallest *p* value, as demonstrated in the univariate analyses of each gene marker as a single indicator in the TCGA dataset (median OS: not reached vs. 1589 days; Appendix A). Furthermore, multivariate Cox analysis demonstrated that *MRPL23*-high expression was an independent risk factor for poor OS (Appendix A). By backward stepwise regression, we established that *MRPL23,* and *MATR3* were the most predictive variables. Of note, based on the UALCAN and TNMplot.com analysis platforms, *MRPL23* levels were markedly elevated in cancer specimens compared with their normal counterparts (Appendix A). Next, we asked whether there was any possible added value of combining *MATR3* and *MRPL23* to the prognostic value of each of the markers alone. Based on the TCGA dataset sourced from the Xena database, two-marker combinations yielded the best separation of the ccRCC patients according to their survival, with a *p*-value of *p* < 0.0001 and patients in *MATR3*-high/*MRPL23*-low group not reaching the median OS, and those in the opposite group experiencing the worst survival (1111 days; Appendix A). Furthermore, a univariate Cox analysis of the combined two-markerset of *MATR3*-low/*MRPL23*-high was associated with poor survival prognosis (HR 4.14, 95%CI 2.71–6.31, *p <* 0.0001; Appendix A) and was a potent independent prognostic factor for ccRCC patients when examined in a multivariate analysis (HR 3.23, 95%CI 2.09–4.99, *p <* 0.0001; Appendix A).

## 4. Discussion

To our knowledge, this is the first study investigating MATR3 expression in ccRCC samples. In the current study, we found that MATR3 was significantly downregulated in ccRCC tissues compared with control tissues in both our in-house and TCGA cohorts from the GEPIA2. We showed that low MATR3 mRNA levels were associated with poor prognosis features such as advanced T-stage and lymph node metastasis. In contrast, the expression status of MATR3 protein was not related to clinicopathological characteristics. Interestingly, we reported that both MATR3 protein and mRNA expression levels were significantly associated with OS of ccRCC patients. Patients with low MATR3 levels had worse OS than those with high MATR3 levels. The association of low MATR3 expression with worse survival is consistent with a tumor suppressive profile; nevertheless, our investigation does not allow us to conclude anything definitive about the character of MATR3 functionality in ccRCC. However, the association of MATR3 mRNA expression with clinicopathological features confirmed this hypothesis. Indeed, Yang et al. previously showed a tumor suppressive function of MATR3 in basal-like breast cancer [9]. The authors demonstrated that forced overexpression of MATR3 promoted apoptotic cell death, suppressed in vitro tumorigenicity, and inhibited epithelial–mesenchymal transition, migration, and invasion. Moreover, low expression levels of MATR3 were related to unfavorable outcomes in breast cancer patients [9]. Likewise, our previous report revealed that low MATR3 protein expression was an independent poor prognostic factor in non-small cell lung cancer [15]. In the current study, low MATR3 expression in our and TCGA cohorts was an adverse prognostic factor in both univariate and multivariate analysis. The opposite relationship was presented by Yang et al. concerning neuroblastoma. The authors of the cited study showed that high expression of MATR3 is associated with poor event-free survival and OS of patients with neuroblastoma from the RNA-seq cohort [16]. These results align with the experimental studies on oral squamous cell carcinoma cells, where suppression of MATR3 by licochalcone H treatment resulted in the induction of cell cycle arrest and apoptosis [17]. Similar cellular effects of MATR3 knockdown have been demonstrated by Kuriyama et al. in in vitro and in vivo models of malignant melanoma [11].

Consistent with previous observations, decreased expression of the DNA double-strand break repair proteins is related to worse survival in cancer patients [18]. In view of the finding that MATR3 is involved in the early response to DNA double-stranded breaks [6], we hypothesize that disruption of this system may be one possible mechanism by which MATR3 halts cancer progression. Our functional enrichment analysis for the MATR3-correlated gene set in ccRCC showed that they are associated with DNA damage bypass, among other processes.

Interestingly, according to the UALCAN database, a majority of MATR3-correlated genes were significantly related to the survival of ccRCC patients, with MRPL23 having the highest prognostic hazard ratio value and the smallest *p*-value in the univariate analyses. High MRPL23 mRNA expression was also an adverse prognostic factor in the multivariate analysis. Therefore, we evaluated whether the combination of MATR3 and MRPL23 could be more informative for prognosis than either protein alone. The two-marker set proved to be a powerful independent prognostic factor and better predicted patient survival than examining each marker individually.

Although the in-house cohort study is limited by sample size and the mRNA data are from the TCGA cohort, it remains an exciting observation that low levels of MATR3 notably correlated with shorter survival rates in patients with ccRCC. Undoubtedly, our findings highlight the role of MATR3 in tumor progression. Furthermore, using bioinformatics analysis, we identified a panel, which includedMATR3 and MRPL23, to predict survival in ccRCC patients. Nevertheless, further investigation is required to confirm these potential correlations.

## 5. Conclusions

In conclusion, MATR3 protein and/or mRNA expression levels are significantly altered in ccRCC and may provide prognostic information as an indicator of overall survival.

## Figures and Tables

**Figure 1 biomedicines-11-00326-f001:**
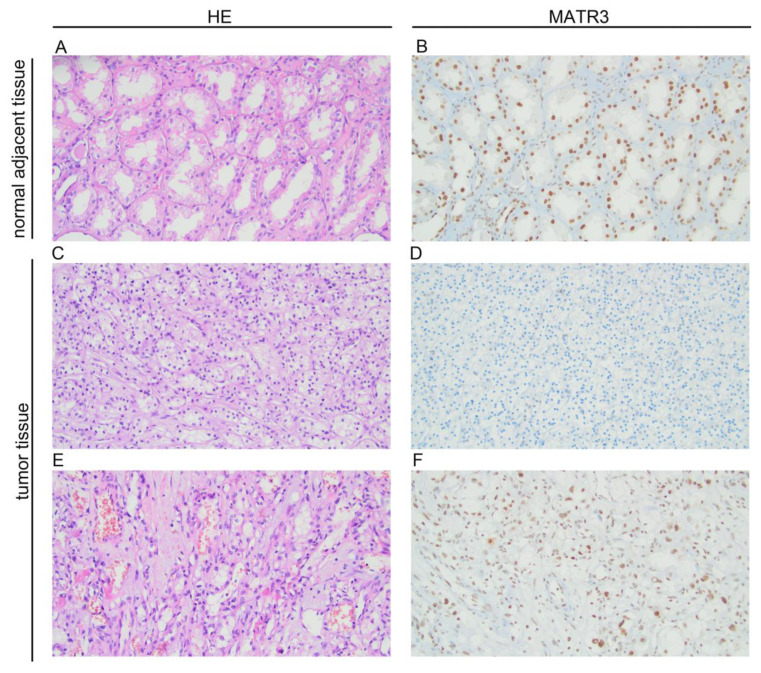
Representative pictures of hematoxylin and eosin staining and immunohistochemical expression of MATR3 in clear cell renal cell carcinoma (ccRCC) and adjacent normal tissues. (**A**) hematoxylin-eosin (HE) staining in adjacent tissue; (**B**) MATR3 staining in adjacent tissue; (**C**) hematoxylin-eosin (HE) staining in ccRCC; (**D**) negative staining for MATR3 in ccRCC; (**E**) hematoxylin-eosin (HE) staining in ccRCC; (**F**) positive staining for MATR3 in ccRCC (primary magnification ×20). Figure prepared using Adobe Photoshop software.

**Figure 2 biomedicines-11-00326-f002:**
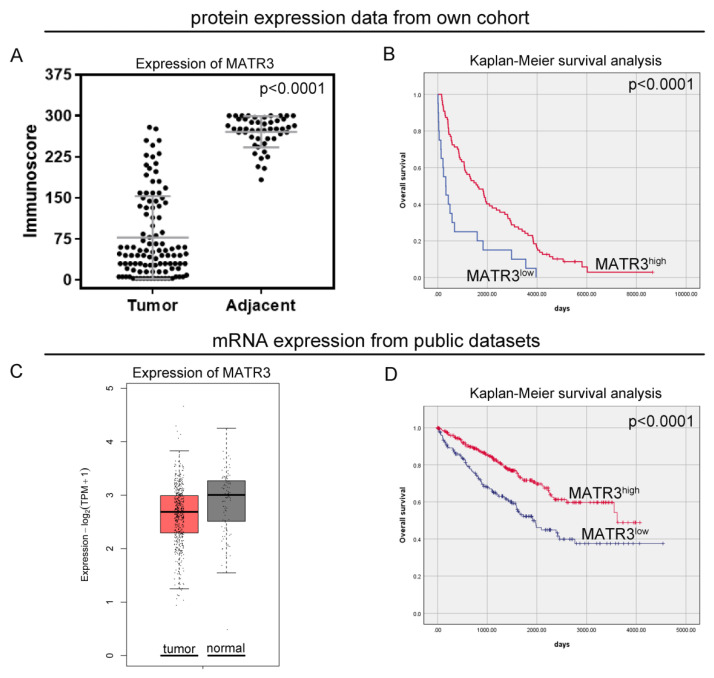
Protein and mRNA expression of MATR3 in ccRCC. Immunohistochemical (**A**) and mRNA (**C**) expression of MATR3 in clear cell renal cell carcinoma (ccRCC) compared to normal tissues. (**B**) Kaplan–Meier survival curves for overall survival of ccRCC patients based on (**B**) MATR3 protein expression and (**D**) MATR3 mRNA expression. Figure prepared using Adobe Photoshop software.

**Figure 3 biomedicines-11-00326-f003:**
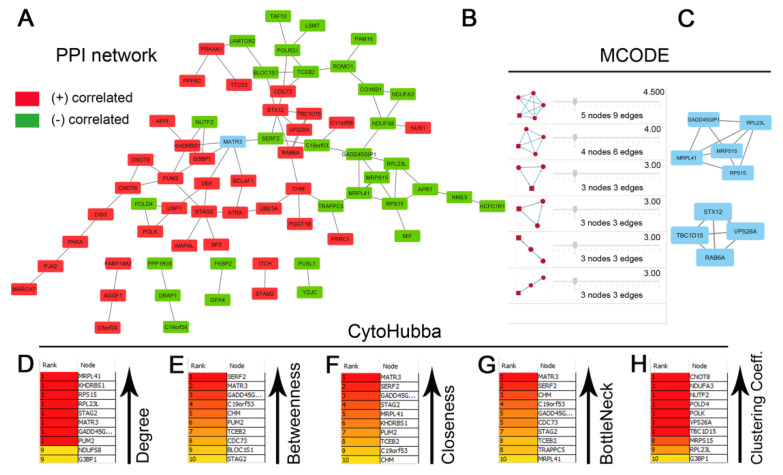
Protein–protein interaction (PPI) network for the top genes positively or negatively correlated with *MATR3* in clear cell renal cell carcinoma (ccRCC) (**A**). The nodes and edges are retrieved from the STRING database and plotted using Cytoscape software. The red nodes represent *MATR3* positively correlated genes, and the green nodes represent *MATR3* negatively correlated genes. *MATR3* is highlighted in blue. Disconnected nodes were hidden. The top 10 hub genes identified by CytoHubba Cytoscape plugin are ranked according to degree centrality (**B**), betweenness centrality (**C**), closeness centrality (**D**), bottleneck (**E**), and clustering coefficient (**F**). The six modules identified in the PPI network using the MCODE Cytoscape plugin (**G**). The top two modules identified from the PPI network could be seen (**H**). Figure prepared using Adobe Photoshop software.

**Figure 4 biomedicines-11-00326-f004:**
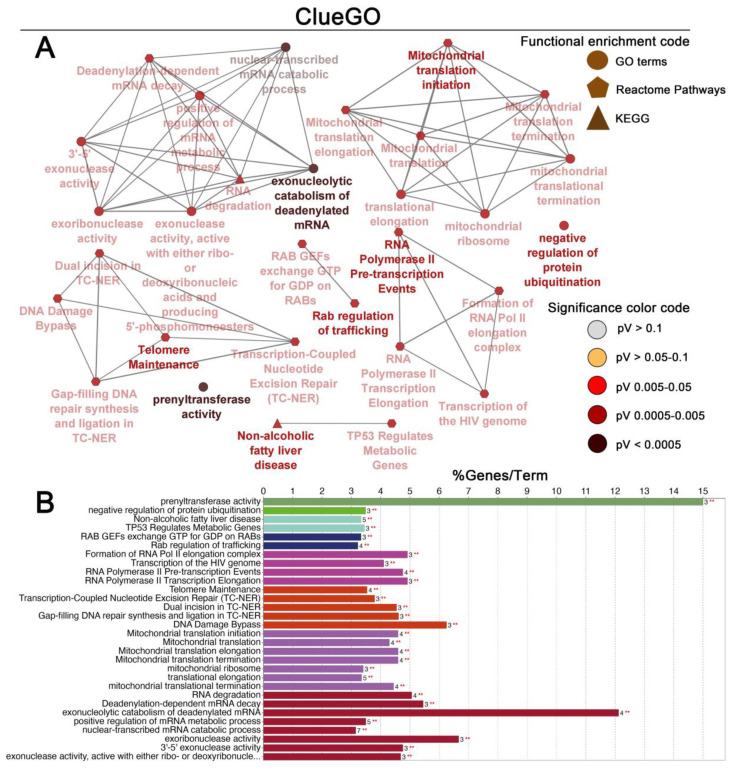
Enrichment of nodes in the PPI network by Gene Ontology (GO) terms, Reactome Pathways and KEGG terms was visualized using the ClueGO plugin in Cytoscape. Schematic illustration of functional enrichment from ClueGO analysis; the enrichment shows only significant pathways (*p*-value ≤ 0.05) (**A**). The bar chart from ClueGO analysis. The bars represent the number of genes associated with the terms (**B**). Figure prepared using Adobe Photoshop software.

**Table 1 biomedicines-11-00326-t001:** MATR3 protein expression and its relationship with clinicopathological features of clear cell renal cell carcinoma (ccRCC) patients in our cohort. “-“ indicates low expression; “+” indicates high expression.

		MATR3		
Variables	Number (%)	+	-	*p*-Value
		*n* = 87	*n* = 20	
**Sex**				
**Females**	32 (29.91)	25 (78.13)	7 (21.88)	0.60
**Males**	75 (70.09)	62 (82.67)	13 (17.33)	
**Age**				
**≤** **65**	63 (58.88)	54 (85.71)	9 (14.29)	0.21
**>65**	44 (41.12)	33 (75.00)	11 (25.00)	
**Grade**				
**G1**	26 (24.30)	20 (76.92)	6 (23.08)	0.32
**G2**	68 (63.55)	58 (85.29)	10 (14.71)	
**G3 andG4**	13 (12.15)	9 (69.23)	4 (30.77)	
**pT status**				
**T1**	31 (28.97)	24 (77.42)	7 (22.58)	0.64
**T2**	30 (28.04)	26 (86.67)	4 (13.33)	
**T3 andT4**	46 (42.99)	37 (80.43)	9 (19.57)	
**cN status**				
**N0**	100 (93.46)	82 (82.00)	18 (18.00)	0.61
**N1**	7 (6.54)	5 (71.43)	2 (28.57)	

**Table 2 biomedicines-11-00326-t002:** Univariate and multivariate analyses of prognostic factors in our own cohort by the Cox proportional hazard model. “-” indicates variable was not included in multivariate analysis.

	Univariate Analysis				Multivariate Analysis			
Variable	HR	95% CI		*p*-Value	HR	95.0% CI		*p*-Value
**MATR3**	2.59	1.57	4.27	<0.0001	2.36	1.41	3.97	0.001
**Sex**	0.60	0.39	0.92	0.02	0.67	0.43	1.04	0.08
**Age**	1.63	1.09	2.43	0.02	1.24	0.81	1.89	0.32
**Grade**	3.48	1.90	6.35	<0.0001	3.83	2.05	7.16	<0.0001
**pT status**	1.12	0.75	1.67	0.58	-	-	-	-
**cN status**	3.36	1.51	7.46	0.003	3.15	1.40	7.09	0.006

**Table 3 biomedicines-11-00326-t003:** *MATR3* mRNA expression and its relationship with clinicopathological features of clear cell renal cell carcinoma (ccRCC) patients in the TCGA cohort. “-” indicates low expression; “+” indicates high expression.

		MATR3		
Variables	Number (%)	+	-	*p*-Value
		*n* = 276	*n* = 199	
**Sex**				
**Females**	163 (34.32)	97 (59.51)	66 (40.29)	0.70
**Males**	312 (65.68)	179 (57.37)	133 (42.63)	
**Age**				
**≤** **60**	239 (50.32)	127 (53.14)	112 (46.86)	0.03
**>60**	236 (49.68)	149 (63.14)	87 (36.86)	
**Grade**				
**G1**	11 (2.32)	9 (81.82)	2 (18.18)	0.35
**G2**	203 (42.74)	118 (58.13)	85 (41.87)	
**G3**	189 (39.79)	105 (55.56)	84 (44.44)	
**G4**	72 (15.16)	44 (61.11)	28 (38.89)	
**pT status**				
**T1**	237 (49.89)	145 (61.18)	92 (38.82)	0.01
**T2**	61 (12.84)	24 (39.34)	37 (60.66)	
**T3**	167 (35.16)	90 (53.89)	77 (46.11)	
**T4**	10 (2.11)	4 (40.00)	6 (60.00)	
**pN status**				
**Nx**	235 (49.47)			
**N0**	225 (47.37)	137 (60.54)	88 (39.11)	0.01
**N1**	15 (3.16)	4 (26.67)	11 (73.33)	
**Stage**				
**I**	234 (49.26)	144 (61.54)	90 (38.46)	0.40
**II**	50 (10.53)	30 (60.00)	20 (40.00)	
**III**	119 (25.05)	64 (53.78)	55 (46.22)	
**IV**	72 (15.16)	38 (52.78)	34 (47.22)	

**Table 4 biomedicines-11-00326-t004:** Univariate and multivariate analyses of prognostic factors in the TCGA cohort by the Cox proportional hazard model. “-” indicates variable was not included in multivariate analysis.

	Univariate Analysis				Multivariate Analysis			
Variable	HR	95% CI		*p*-Value	HR	95.0% CI		*p*-Value
**MATR3**	1.97	1.44	2.70	<0.0001	2.01	1.46	2.75	<0.0001
**Sex**	0.95	0.68	1.31	0.74	-	-	-	-
**Age**	1.06	0.77	1.44	0.74	-	-	-	-
**Grade**	1.36	0.98	1.87	0.06	1.17	0.84	1.61	0.35
**pT**	3.19	2.31	4.39	<0.0001	-	-	-	-
**pN**	3.65	1.93	6.90	<0.0001	-	-	-	-
**TNM stage**	3.61	2.59	5.02	<0.0001	3.60	2.58	5.02	<0.0001

## Data Availability

The datasets generated during and/or analyzed during the current study are available from the corresponding author on reasonable request.

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
