# Peer review of "Low Expression of MATR3 Is Associated with Poor Survival in Clear Cell Renal Cell Carcinoma"

_biomedicines, 2023, doi:10.3390/biomedicines11020326_

Round 1
Reviewer 1 Report
The paper continues to lack verification that MATR3 is functional in RCC. Both reviewers have requested fairly straight-forward in vitro experiments, and these experiments have not been performed. The protein expression has been measured on a small cohort and not validated independently.
Author Response
We know that our results should be interpreted cautiously because of these limitations. This study needs to be validated - as we emphasized in the discussion section. Nevertheless, we made sure that our investigation was conducted reliably. Our research, as others in the cancer field, is intended to add a proverbial ‘puzzle element’ to global efforts toward understanding the biology of ccRCC and finding new prognostic markers. Unfortunately, it is not possible to conduct additional analysis in the current proposal. We will continue to explore this topic and will certainly take your advice in designing future projects.

Reviewer 2 Report
Dear authors,
After the review process, I have several comments: you should add numerical data in the abstract; you should include in the figures legend how it was realized; limitations of the study should be included in section 4.
Best regards!
Author Response
Dear Reviewer,
We are submitting the manuscript's revised version for you to review for publication in Biomedicines. We want to thank you for the time you spent reviewing the manuscript and your thoughtful comments that helped to improve it. The manuscript has been revised according to suggestions. All co-authors have agreed to the revisions. Below, we have provided specific responses.
Yours sincerely,
Justyna Durślewicz
Point 1: You should add numerical data in the abstract;
Response: According to a suggestion, the authors added numerical data in the abstract.
Point 2: You should include in the figures legend how it was realized;
Response: The authors have included ifnromation regarding the preparation of the figures.
Point 3: Limitations of the study should be included in section 4.
Response: Thank you for your comment; the study's limitations have been moved to Section 4.

Reviewer 3 Report
Comments:
1. Any data from TCGA and other databases to support the current conclusion?
2. On Table 1, do authors have patients' info on blood pressure level, BML, glucose level, etc?
3. Since MRPL23 is strongly associated with MATR3, do authors have any direct data between MRPL23 and MARP3 such as RT-PCR and Western blot analysis?
Author Response
Dear Reviewer,
We are submitting the manuscript's revised version for you to review for publication in Biomedicines. We want to thank you for the time you spent reviewing the manuscript and your thoughtful comments that helped to improve it. All co-authors have agreed to the revisions. Below, we have provided specific responses.
Yours sincerely,
Justyna Durślewicz
Point 1. Any data from TCGA and other databases to support the current conclusion?
Response: Our work aimed to evaluate MATR3 immunohistochemical staining and RNA-seq data from publicly available datasets, including TCGA and GEPIA2. Information on publicly available databases is described in Section 2.4. Analysis of the Database.
2.4. Analysis of the Database
Apart from the protein level, the mRNA expression levels of the targeted genes were compared using public data. MATR3 mRNA expression between ccRCC and “match TCGA normal and GTEx data” was compared using the public web server GEPIA2 (Gene Expression Profiling Interactive Analysis; http://gepia.cancer-pku.cn/ accessed on 11 April 2022). Furthermore, gene expression, clinicopathological, and survival data for a cohort of 475 ccRCC patients were obtained from UCSC Xena Browser (http://xena.ucsc.edu/ accessed on 8 April 2022). RNA-sequencing data for MATR3 were normalized via DESeq2 normalization. The detailed characterization of the TCGA cohort is shown in Supplementary Table 4. The data were categorized into low-level (<10.77) and high-level (≥10.77) MATR3 expression groups according to the cutoff point established using the Evaluate Cutpoints software[18].
Point 2. On Table 1, do authors have patients' info on blood pressure level, BML, glucose level, etc?
Response: Thank you for your comment. We know that the analysis involving additional parameters such as patients’ blood pressure level, BML, glucose level, etc., data would be a valuable complement to our results. Unfortunately, these data are unavailable in the context of our study in both our and TCGA cohorts. Tables 1 and 3 focus exclusively on clinicopathological features.
Point 3. Since MRPL23 is strongly associated with MATR3, do authors have any direct data between MRPL23 and MARP3 such as RT-PCR and Western blot analysis?
Response: The data on the relationship between MATR3 and MRPL23 are obtained through public database analyses. This report is preliminary and is the first to set the stage for further and more detailed studies on the combination of MATR3 and MRPL23 in ccRCC. We agree with the reviewer's comment that the results of RT-PCR and Western Blot analyses would be very powerful. Unfortunately, it is impossible to carry out these analyses in the current proposal. We will continue looking into this topic and will definitely use your advice to design the following projects.

Round 2
Reviewer 3 Report
No more comments